# True Stress Theory of Matrix in A Composite: A Topical Review

**DOI:** 10.3390/ma16020774

**Published:** 2023-01-12

**Authors:** Zheng-Ming Huang

**Affiliations:** School of Aerospace Engineering and Applied Mechanics, Key Laboratory of the Ministry of Education for Advanced Civil Engineering Materials, Tongji University, 1239 Sipping Road, Shanghai 200092, China; huangzm@tongji.edu.cn; Tel.: +86-021-65985373

**Keywords:** composites, true stress, stress concentrations, failures

## Abstract

**Highlights:**

Non-uniform matrix stresses in composites cannot be assessed against matrix strengths obtained in uniform stress states.

The non-uniform matrix stresses must be converted into equivalent uniform ones called true stresses before estimating any matrix-dominated composite failure.

Conversion is achieved through matrix true stress theory summarized in this paper.

Selected applications for resolving several challenging problems in composite failures are highlighted.

**Abstract:**

Whereas mechanics theories for isotropic materials are almost matured, only linear elastic theories for composites were essentially established. This is because only homogenized or approximated stresses are obtainable for a composite. Its mechanical properties must be estimated on a true stress level. According to Eshelby, the true stresses of the fiber are the same as its homogenized counterparts. The true stress theory for the matrix was systematically established by the author, and is reviewed and summarized in the paper. An Excel table-based program for calculating all of the possible true stress components is provided as a supplement for the reader to download. As most composite failures are caused by matrix failures, the true stress theory plays a predominant role in estimating the composite properties outside a linear elastic range. Some challenging composite failures were resolved upon the matrix true stresses, and are highlighted in the paper.

## 1. Introduction

The objectives investigated by solid mechanics can be classified into isotropic and anisotropic. The mechanics theories of isotropic materials are almost matured. Namely, almost every mechanics problem of an isotropic material is able to be well dealt with using existing theories. In contrast, the matured mechanics theories for anisotropic or composite materials are limited to linear elasticity only. All of the other mechanics problems of the composites, such as nonlinear constitutive relations or deformations, and failure and strength predictions, are overall not able to be resolved satisfactorily.

A fundamental reason is that only homogenized stresses are obtainable for a composite by the current theories, and the resulting internal stresses of the fiber and matrix are the homogenized values as well. They must be converted into true stresses before a composite property can be analyzed, as is the case for an elastic property of the composite as well. Only because the homogenized and the true stresses are in a linear elastic range are the resulting composite elastic properties predicted on both of them the same, giving rise to a false impression that an elastic property of the composite can be independent of its true stresses. A schematic diagram to illustrate this concept is shown in Figure 1.

The stress field of the fiber (standing for the reinforcement subsequently) in a continuous fiber composite is uniform [1,2,3]. Its true and homogenized stresses are the same. The true stresses of the fiber in a short fiber/particle composite are immaterial, since the fiber/particle is generally linear elastic up to rupture and less possible to fail before a matrix failure. Furthermore, a short fiber can be sufficiently well considered as an ellipsoid of a finite length. According to Eshelby [1], the stress field in the ellipsoid is uniform no matter what kind of load is applied to the composite. Thus, the bottleneck for advancing mechanics of composites is to evaluate the true stresses of the matrices in the composites.

We found a true stress of the matrix is determined by multiplying its homogenized counterpart with a stress concentration factor (SCF) of the matrix in the composite [4,5]. A plate with a hole generates a stress concentration at the hole edge. When the hole is filled with a fiber of different properties, a stress concentration occurs as well. However, such an SCF cannot be defined through a classical method any more. Otherwise, the resulting SCF would be infinite if an interface crack between the fiber and matrix occurred, as the matrix stress field at the crack tip is singular. Even though the interface is in perfect bonding up to failure, the SCF given by the classical method is still less pertinent since a predicted composite failure incorporated with it is still far away from a measured counterpart. We succeeded in achieving the new definition [4,5], and almost all of the possible SCF formulae were derived [4,5,6,7,8,9,10,11]. The matrix true stresses were then determined accordingly.

It is evident that the load share of the fiber attains the highest when a continuous fiber composite is subjected to a longitudinal load, and thus a fiber fracture can occur most possibly. Even in such a case, however, a fiber splitting failure is often seen at a longitudinal tension (Figure 2a), and a fiber kinking failure occurs most frequently at a longitudinal compression (Figure 2b [12]), both of which are caused by a matrix shear failure [13,14]. Due to an inevitable fiber misalignment in a composite fabrication, a longitudinal load generates a shear stress component in the misaligned coordinate system (Figure 2c). The axial stress component will cause the fiber to possibly fail, whereas the shear component can bring a failure to the matrix. Apparently, both the splitting and kinking failure modes indicate that the matrix shear failures occur earlier than a fiber fracture. Otherwise, the composite failure surface would be perpendicular to the loading direction.

The above evidences demonstrate that almost all of the composite failures are essentially caused by matrix failures. Without the true stress theory, none of the matrix failures can be analyzed properly. An Excel table-based program to calculate almost all of the possible matrix SCFs derived by the author et al. [4,5,6,7,8,9,10,11] was worked out, and is attached as a supplement with this paper for a reader to download (Appendix A). One can automatically obtain the desired results by running the program with input of the monolithic fiber and matrix property data and the fiber geometric parameters. For a continuous fiber composite, only the fiber volume fraction is required. For a discontinuous fiber composite, two other geometric parameters, i.e., the fiber aspect ratio and the fiber length ratio, should be provided as well. Using the true stress concept, a number of long-standing and challenging problems in composite failures are resolved satisfactorily and easily. They include a necessary and sufficient condition for a fiber and matrix interface debonding to occur given a composite subjected to an arbitrary load [8], a fiber splitting [13] or kinking [14] failure versus a fiber fracture, and a large shear deformation of the composite induced from relative slippage displacements between debonded fiber and matrix interfaces [15]. All of these failures are analyzed with no iteration. All of these failures are analyzed using fiber and matrix properties plus limited composite data measured independently and following existing standards. The matrix SCF formulae, together with their applications to the analyses of the challenging composite failures, are briefly reviewed and summarized in the paper.

## 2. Homoginized Stresses

In mechanics of continuum media, a stress at a media point is defined as the average of those on an infinitesimally small unit element containing the point through
(1)$$\sigma_{i}=\frac{1}{V^{\prime }}\int_{V^{\prime }}{\tilde{\sigma }}_{i}dV^{\prime },V^{\prime }\to 0\;\;(i=11,22,33,23,13,12)$$
where ${\tilde{\sigma }}_{i}$ is called a point-wise stress. For a composite, however, its unit element (called a representative volume element or RVE) cannot be infinitesimal, since both the fiber and matrix must be contained in it. Equation (1) becomes
(2)$$\sigma_{i}=\frac{1}{V^{\prime }}\int_{V^{\prime }}{\tilde{\sigma }}_{i}dV=\frac{V_{f}^{\prime }}{V^{\prime }}(\frac{1}{V_{f}^{\prime }}\int_{V_{f}^{\prime }}{\tilde{\sigma }}_{i}dV)+\frac{V_{m}^{\prime }}{V^{\prime }}(\frac{1}{V_{m}^{\prime }}\int_{V_{m}^{\prime }}{\tilde{\sigma }}_{i}dV)=V_{f}\sigma_{i}^{f}+V_{m}\sigma_{i}^{m}$$
where *V*′ =$V_{f}^{\prime }$+$V_{m}^{\prime }$. $V_{f}^{\prime }$ and $V_{m}^{\prime }$ are the fiber and matrix volumes in the RVE, respectively, and *V* is a volume fraction. Hence, a stress or strain in a composite is a homogenized or approximated quantity by nature. No exact value of it is available, although, for a constituent, e.g., matrix, both the point-wise and the homogenized stresses, ${\tilde{\sigma }}_{i}^{m}$ and $\sigma_{i}^{m}$, are able to be determined.

Mechanics is a rigorous science. Only an infinitesimally small element is mathematically rigorous enough to denote a material point, regardless of any geometric shape for the element. Although an RVE for a composite cannot be infinitesimal, its purpose is to define a stress for the composite. It is equivalent to an infinitesimally small element for defining a stress in an isotropic solid. Therefore, an RVE cannot be literally considered as a representative of the entire composite, since at different points the composite microstructure and the stresses can be different.

From the very fundamental definition, it is seen that the smaller the chosen RVE is, the more accurate the homogenized stress will be in general. Indeed, this was verified through a comparison study with experiments in [16]. One of the smallest RVEs for a continuous fiber composite is shown in Figure 3a, whereas that for a short fiber composite is indicated in Figure 3b, both of which are transversely isotropic and resemble a unidirectional (UD) and a uniaxially aligned short fiber composite, respectively. It is noticed that when the fiber aspect ratio, *l*/*a*, is close to 1, Figure 3b becomes an RVE of a particle composite.

Using a bridging equation, $\left\{\sigma_{i}^{m}\right\}$=$\left[a_{ij}^{}]\{\sigma_{j}^{f}\right\}$, to correlate the homogenized stresses of the matrix with those of the fiber, one obtains [17]
(3)$$\left\{\sigma_{i}^{f}\}={\left(V_{f}[I]+V_{m}[a_{ij}]\right)}^{-1}\{\sigma_{j}\right\}$$
(4)$$\left\{\sigma_{i}^{m}\}=[a_{ij}]{\left(V_{f}[I]+V_{m}[a_{ij}]\right)}^{-1}\{\sigma_{j}\right\}$$
(5)$$[S_{ij}]=(V_{f}[S_{ij}^{f}]+V_{m}[S_{ij}^{m}][a_{ij}]){\left(V_{f}[I]+V_{m}[a_{ij}]\right)}^{-1}$$

Equation (5) implies that the compliance tensor [*S_ij_*] in a constitutive relation, {*ε_i_*} = [*S_ij_*]{*σ_j_*}, of the composite is equivalent to its bridging tensor [*a_ij_*], in which $\left[S_{ij}^{f}\right]$ and $\left[S_{ij}^{m}\right]$ are, respectively, the compliance tensors of the fiber and matrix known in advance.

There are a number of micromechanics models, either analytical or numerical, which can be used to determine an elastic bridging tensor of the composite. The bridging model established by this author [17,18] has several unique features. First, it is the only available constitutive theory that is consistent in calculating the internal stresses [16]. Let [*S_ij_*]_2D_, [*a_ij_*]_2D_, ${\left[S_{ij}^{f}\right]}_{2D}$, and ${\left[S_{ij}^{m}\right]}_{2D}$ denote the 2D (two-dimensional) quantities deteriorated from the corresponding 3D ones, [*S_ij_*], [*a_ij_*], $\left[S_{ij}^{f}\right]$, and $\left[S_{ij}^{m}\right]$. Let the composite be subjected to a planar stress state, {*σ_j_*}_2D_ = {*σ*_11_,*σ*_22_,*σ*_12_}*^T^*. The internal stresses of the fiber and matrix can be calculated from Equations (3) and (4) in either the 2D or the 3D formulae, with a 3D stress vector given by, e.g., {*σ_j_*} = {*σ*_11_,*σ*_22_,0,0,0,*σ*_12_}*^T^*. If the resulting internal stresses from the two sets of formulae are exactly the same, the constitutive theory is consistent in the internal stress calculation. A necessary and sufficient condition for a constitutive theory to be consistent is that its bridging tensor is upper-triangular. Only the bridging tensor of the bridging model is always upper-triangular, whereas all of the others known yet are not [16]. For a non-consistent theory, its 3D formulae have to be used to calculate the internal stresses of the fiber and matrix more accurately. The calculation amount is at least one order higher than that by using the 2D ones. Second, it is a unified analytical constitutive model for a composite reinforced with either continuous fibers, or short fibers, or particles, no matter the constituent, e.g., matrix, undergoes an elastic or plastic deformation [17,18,19,20]. Third, it is overall more accurate than most other micromechanics including numerical models [21,22,23].

When both the fiber and matrix are in elastic deformations, the homogenized internal stresses by the bridging model are expressed as [7]
(6)$$\sigma_{11}^{f}=\frac{\sigma_{11}^{}}{V_{f}+V_{m}a_{11}}-\frac{V_{m}a_{12}(\sigma_{22}^{}+\sigma_{33}^{})}{\left(V_{f}+V_{m}a_{11})(V_{f}+V_{m}a_{22}\right)}$$
(7)$$\sigma_{11}^{m}=\frac{a_{11}\sigma_{11}^{}}{V_{f}+V_{m}a_{11}}+\frac{V_{f}a_{12}(\sigma_{22}^{}+\sigma_{33}^{})}{\left(V_{f}+V_{m}a_{11})(V_{f}+V_{m}a_{22}\right)}$$
(8)$$\sigma_{ij}^{f}=\frac{\sigma_{ij}^{}}{V_{f}+V_{m}a_{22}},\;ij=22,33,23$$


(9)
$$\sigma_{ij}^{m}=\frac{a_{22}\sigma_{ij}^{}}{V_{f}+V_{m}a_{22}},\;ij=22,33,23$$



(10)
$$\sigma_{ij}^{f}=\frac{\sigma_{ij}^{}}{V_{f}+V_{m}a_{66}},\;ij=12,13$$



(11)
$$\sigma_{ij}^{m}=\frac{a_{66}\sigma_{ij}^{}}{V_{f}+V_{m}a_{66}},\;ij=12,13$$


In Equations (6)–(11), {*σ*_11_,*σ*_22_,*σ*_33_,*σ*_23_,*σ*_13_,*σ*_12_} are any loads applied on the RVE or a UD/uniaxially aligned short fiber composite. Explicit expressions for the nonzero bridging tensor elements are as follows [17,18]:(12)$$a_{11}=\left\{\begin{array}{l} E^{m}/E_{11}^{f},for_{}\;a_{}\;continuous_{}\;fiber_{}\;composite\\ \frac{V_{f}E^{m}}{V_{m}E_{11}^{f}}\frac{\sigma_{11}L-E_{11}^{f}[\varepsilon_{L}^{1}l+\varepsilon_{L}^{2}(L-l)]}{E_{}^{m}[\varepsilon_{L}^{1}l+\varepsilon_{L}^{2}(L-l)]-\sigma_{11}L},for_{}\;a_{}\;short_{}\;fiber/particle_{}\;composite \end{array}\right.$$
(13)$$a_{22}=a_{33}=a_{44}=0.3+0.7E^{m}/E_{22}^{f}$$
(14)$$a_{55}=a_{66}=0.3+0.7G^{m}/G_{12}^{f}$$
(15)$$a_{12}=a_{13}=\frac{S_{12}^{f}-S_{12}^{m}}{S_{11}^{f}-S_{11}^{m}}(a_{11}-a_{22})=\frac{E_{11}^{f}\nu^{m}-E^{m}\nu_{12}^{f}}{E_{11}^{f}-E^{m}}(a_{22}-a_{11})$$

$E_{11}^{f}$, $E_{22}^{f}$, $G_{12}^{f}$ and $\nu_{12}^{f}$ are, respectively, the longitudinal, transverse, in-plane shear moduli and longitudinal Poisson’s ratio of the fiber. *E^m^*, *G^m^*, and *ν^m^* are the matrix tensile and shear moduli and Poisson’s ratio; $\varepsilon_{L}^{1}$ and $\varepsilon_{L}^{2}$ are the homogenized uniaxial strains of the regions Ω_1_ and Ω_2_ in Figure 3b, which are determined as follows [18]:
(16)$$\left[\begin{array}{cccc} 0 & 0 & \frac{l}{L}[gE_{11}^{f}+(1-g)E^{m}] & (1-l/L)E^{m}\\ \frac{2\nu^{m}E^{m}}{\left(1+\nu^{m})(1-2\nu^{m}\right)} & \frac{2g\nu^{m}E^{m}}{\left(1+\nu^{m})(1-2\nu^{m}\right)} & 0 & \frac{(1-\nu^{m})E^{m}}{\left(1+\nu^{m})(1-2\nu^{m}\right)}\\ \frac{E^{m}[g\nu_{12}^{f}+\nu^{m}(1-g)]}{0.5(1+\nu^{m})(1-2\nu^{m})} & -\frac{2g\nu_{12}^{f}E^{m}}{1+\nu^{m}} & k_{1} & 0\\ k_{2} & k_{3} & k_{4} & 0 \end{array}\right]\left\{\begin{array}{l} C_{1}\\ \frac{C_{2}}{a^{2}}\\ \varepsilon_{L}^{1}\\ \varepsilon_{L}^{2} \end{array}\right\}=\left\{\begin{array}{l} \sigma_{11}\\ \sigma_{11}\\ \sigma_{11}\\ 0 \end{array}\right\}$$
(17)$$k_{1}=\frac{2g\nu^{m}\nu_{12}^{f}+(1-\nu^{m})(1-g)}{\left(1+\nu^{m})(1-2\nu^{m}\right)}E^{m}+gE_{11}^{f},g=(a/b{)}^{2}$$
(18)$$k_{2}=E_{11}^{f}\frac{E^{m}\nu_{12}^{f}(1+\nu_{23}^{f})+\nu_{12}^{f}E_{22}^{f}-(1+2\nu^{m})\nu^{m}\nu_{12}^{f}E_{22}^{f}}{\left(1+\nu^{m})(1-2\nu^{m})[E_{11}^{f}-E_{22}^{f}{\left(\nu_{12}^{f}\right)}^{2}\right]}-\frac{2E^{m}\nu_{12}^{f}}{\left(1+\nu^{m})(1-2\nu^{m}\right)}$$
(19)$$k_{3}=E_{11}^{f}\frac{\nu_{12}^{f}(1+\nu^{m})E_{22}^{f}-(1+\nu_{23}^{f})E^{m}\nu_{12}^{f}}{\left(1+\nu^{m})[E_{11}^{f}-E_{22}^{f}{\left(\nu_{12}^{f}\right)}^{2}\right]}+\frac{2E^{m}\nu_{12}^{f}}{1+\nu^{m}}$$
(20)$$k_{4}=E_{11}^{f}\frac{E_{22}^{f}{\left(\nu_{12}^{f}\right)}^{2}(1+\nu^{m})(1-2\nu^{m})+E^{m}\nu^{m}\nu_{12}^{f}(1+\nu_{23}^{f})}{\left(1+\nu^{m})(1-2\nu^{m})[E_{11}^{f}-E_{22}^{f}{\left(\nu_{12}^{f}\right)}^{2}\right]}-\frac{2\nu^{m}E^{m}\nu_{12}^{f}}{\left(1+\nu^{m})(1-2\nu^{m}\right)}$$

$\nu_{23}^{f}$ is the fiber transverse Poisson’s ratio. Without loss of any generality, one can set *σ*_11_ = 1 on the right hand side of Equations (12) and (16). When the fiber aspect ratio *l*/*a* → ∞ or its length ratio *l*/*L* → 1 (see Figure 3b for definition), the expression of *a*_11_ for a short fiber composite in Equation (12) is near to that for a continuous one. The five elastic moduli of the UD or uniaxially aligned short fiber composite are given by [7]
(21)$$E_{11}=\frac{(V_{f}+V_{m}a_{11})E_{11}^{f}E_{}^{m}}{V_{f}E_{}^{m}+V_{m}a_{11}E_{11}^{f}}$$
(22)$$\nu_{12}=\frac{E_{11}(V_{f}\nu_{12}^{f}E_{}^{m}+V_{m}a_{11}\nu_{}^{m}E_{11}^{f})}{E_{11}^{f}E_{}^{m}(V_{f}+V_{m}a_{11})}$$
(23)$$E_{22}=\frac{\left(V_{f}+V_{m}a_{11})(V_{f}+V_{m}a_{22}\right)}{(V_{f}+V_{m}a_{11})(V_{f}S_{22}^{f}+a_{22}V_{m}S_{22}^{m})+V_{f}V_{m}(S_{21}^{m}-S_{21}^{f})a_{12}}$$
(24)$$G_{12}=\frac{G_{12}^{f}G^{m}(V_{f}+V_{m}a_{66})}{V_{f}G^{m}+V_{m}G_{12}^{f}a_{66}}$$
(25)$$G_{23}=\frac{G_{23}^{f}G^{m}(V_{f}+V_{m}a_{22})}{V_{f}G^{m}+V_{m}G_{23}^{f}a_{22}}.$$

It is noted that among the three geometric parameters used to define the RVE geometry of a short fiber composite (Figure 3b), *V_f_*, *l*/*a* and *l*/*L*, only *V_f_* and *l*/*a* are measurable in advance. The length ratio *l*/*L* is not directly measurable. An empirical expression for it is proposed in [19], which reads
(26)$$\frac{l}{L}={\left[\sqrt{V_{f}}\frac{l}{a}\tan\left\{\arctan(a/l)-c(l-a)\frac{\arctan(a/l)-\arctan(a/(l\sqrt{V_{f}}))}{c(l-a)+a}\right\}\right]}^{2/3}$$
where *c* = 0.03 is an empirical parameter [19].

For any structure that is not a UD, nor a uniaxially aligned short fiber composite, as schematically shown in Figure 4, a subdivision of it into finite elements is necessary. With a further application of a lamination theory [7], the load share by each lamina layer in the laminated element is obtained. Then, the lamina layer is cut in an arbitrary way into a series of sheets, each of which contains at most one straight fiber yarn segment. Such a sheet is considered as an RVE of either Figure 3a or Figure 3b in its local coordinate system, with possibly a different *V_f_*. Some sheets may even contain no fiber. Namely, they are pure matrix unit elements. The bridging model formulae summarized above are then applied to determine the homogenized internal stresses in the fiber and matrix, as well as the compliance tensor of the RVE. The homogenized internal stresses are then converted into true values as per the methods summarized in the subsequent section. An assemblage in terms of, e.g., an iso-strain or iso-stress scheme [24,25], from all of the RVEs gives rise to the overall internal true stresses in the fiber and matrix and the compliance tensor of the lamina layer.

## 3. True Stress

### 3.1. Background

Although the elastic properties of a composite are calculated by the bridging model formulae, Equations (21)–(25), correlate well with the experiments [21,22,23], and a predicted composite failure based on the internal stresses by Equations (6)–(11) can be very much different from reality. For instance, the transverse tensile strength of a UD composite thus obtained can be more than five times bigger than the measured counterpart [16,20]. A similar conclusion on the poor agreement is applicable to any other micromechanics theory.

This demonstrates that the homogenized internal stresses must be converted into true values before a failure detection can be made. The fiber stress field in Figure 3a is uniform, and the fiber true and homogenized stresses are the same; the fiber in Figure 3b generally does not fail, and its true stresses are immaterial. Further, a short fiber can be well regarded as an ellipsoid of a finite length, and its stresses are uniform [1]. We only need to evaluate the true stress of the matrix, which is equal to its homogenized counterpart multiplied by a factor. This factor was still called an SCF by the author [4,5], as it has all the physical characteristics of a classical SCF. Namely, it is dimensionless, defined by a stress ratio, relevant to a material failure, and dependent on a loading manner.

### 3.2. Definition

As an SCF of the matrix in a composite can no longer be defined by a classical method, which is given as a point-wise stress divided by an overall applied one, an averaging of the stresses in both the numerator and denominator must be made. The “point-wise stress” in the numerator has a geometrical characteristic of zero dimensions (0D), whereas the “overall applied stress” in the denominator is in fact a surface-averaged quantity (2D) with respect to the surface where the load is being applied. By similarity, the new definition must be given by 1D over 3D geometries, i.e., a line-averaged stress divided by a volume-averaged one. Further, the line averaging in the numerator should be along the outward normal of the failure surface of the composite under the given load, from the fiber cylinder to the matrix one in the RVE [4,5].

### 3.3. Matrix SCFs with Perfect Interface Bonding [4,5,6,7,8]

The matrix SCFs under transverse tension, transverse compression, transverse shear, and longitudinal shear, $K_{22}^{t}$, $K_{22}^{c}$, *K*_23_, and *K*_12_, are derived as
(27)$$K_{22}^{t}=\left[1+\frac{\sqrt{V_{f}}}{2}A+\frac{\sqrt{V_{f}}}{2}(3-V_{f}-\sqrt{V_{f}})B\right]\frac{(V_{f}+0.3V_{m})E_{22}^{f}+0.7V_{m}E^{m}}{0.3E_{22}^{f}+0.7E^{m}}$$
$$K_{22}^{c}=\left\{1-\frac{\sqrt{V_{f}}}{2}A\frac{\sigma_{u,c}^{m}-\sigma_{u,t}^{m}}{2\sigma_{u,c}^{m}}+\frac{B}{2(1-\sqrt{V_{f})}}\left[-V_{f}^{2}\left(1-2{\left(\frac{\sigma_{u,c}^{m}-\sigma_{u,t}^{m}}{2\sigma_{u,c}^{m}}\right)}^{2}\right)+\right.\right.$$
$$\frac{(\sigma_{u,c}^{m}+\sigma_{u,t}^{m})V_{f}}{\sigma_{u,c}^{m}}\left(1+\frac{\sigma_{u,c}^{m}-\sigma_{u,t}^{m}}{\sigma_{u,c}^{m}}\right)-\sqrt{V_{f}}\left.\left.\left(\frac{\sigma_{u,c}^{m}-\sigma_{u,t}^{m}}{\sigma_{u,c}^{m}}+1-2{\left(\frac{\sigma_{u,c}^{m}-\sigma_{u,t}^{m}}{2\sigma_{u,c}^{m}}\right)}^{2}\right)\right]\right\}\times$$
(28)$$\frac{(V_{f}+0.3V_{m})E_{22}^{f}+0.7V_{m}E^{m}}{0.3E_{22}^{f}+0.7E^{m}}$$
(29)$$K_{23}=2\sigma_{u,s}^{m}\sqrt{\frac{K_{22}^{t}K_{22}^{c}}{\sigma_{u,t}^{m}\sigma_{u,c}^{m}}}$$
(30)$$K_{12}=\left[1-V_{f}\frac{G_{12}^{f}-G^{m}}{G_{12}^{f}+G^{m}}\{W(V_{f})-\frac{1}{3}\}\right]\frac{(V_{f}+0.3V_{m})G_{12}^{f}+0.7V_{m}G^{m}}{0.3G_{12}^{f}+0.7G^{m}}$$
(31)$$A=\frac{2E_{22}^{f}E^{m}{\left(\nu_{12}^{f}\right)}^{2}+E_{11}^{f}\{E^{m}(\nu_{23}^{f}-1)-E_{22}^{f}[2{\left(\nu^{m}\right)}^{2}+\nu^{m}-1]\}}{E_{11}^{f}[E_{22}^{f}+E^{m}(1-\nu_{23}^{f})+E_{22}^{f}\nu^{m}]-2E_{22}^{f}E^{m}{\left(\nu_{12}^{f}\right)}^{2}}$$
(32)$$B=\frac{E^{m}(1+\nu_{23}^{f})-E_{22}^{f}(1+\nu^{m})}{E_{22}^{f}[\nu^{m}+4{\left(\nu^{m}\right)}^{2}-3]-E^{m}(1+\nu_{23}^{f})}$$
(33)$$W(V_{f})=\pi \sqrt{V_{f}}[\frac{1}{4V_{f}}-\frac{1}{32}-\frac{1}{256}V_{f}-\frac{5}{4096}V_{f}{}^{2}]$$

$\sigma_{u,c}^{m}$ and $\sigma_{u,s}^{m}$ are the matrix compressive and shear strengths, respectively.

### 3.4. Matrix SCFs after Interface Debonding [8,9]

The in-plane shear and transverse tensile SCFs after the interface debonding are
K^12=Gm(1+Vf)Gm+G12f+G12fGm+G12fπVf2+Vf1−VfIm1Vf−1−2i1Vf−1
$$+\ln\left(\sqrt{1-V_{f}-2i\sqrt{1-V_{f}}}-\sqrt{V_{f}}\right)-i\sqrt{1-\frac{1}{V_{f}}}-\ln(\sqrt{V_{f}-1}-\sqrt{V_{f}})$$
$$+\ln\left(\sqrt{\frac{1}{V_{f}}-1-2i\sqrt{\frac{1}{V_{f}}-1}}-\sqrt{\frac{1}{V_{f}}-1}+i\right)-\sqrt{1+\frac{V_{f}}{1-2V_{f}-2i\sqrt{V_{f}-{\left(V_{f}\right)}^{2}}}}$$
(34)$$\left.\left.\left.-\ln\left(\sqrt{1-\frac{1}{V_{f}}}+\frac{i}{\sqrt{V_{f}}}\right)\right)\right]\right\}\frac{V_{f}G_{12}^{f}+V_{m}(0.3G_{12}^{f}+0.7G^{m})}{0.3G_{12}^{f}+0.7G^{m}}$$
K^22t=Ree−2iψM(beiψ)(a2/b−b)−e−iψN2−N1(a2be−iψ)
(35)$$\left.+e^{-i\psi }(2+e^{-2i\psi })[N(be^{i\psi })-N_{3}]\right\}\frac{(V_{f}+0.3V_{m})E_{22}^{f}+0.7V_{m}E^{m}}{2(b-a)(0.3E_{22}^{f}+0.7E^{m})}$$
(36)$$N(z)=Fz+\frac{a^{2}k}{z}-{\left(z-ae^{i\psi }\right)}^{0.5+i\lambda }{\left(z-ae^{-i\psi }\right)}^{0.5-i\lambda }[(F-0.5)-\frac{D}{a^{2}z}]$$
(37)$$N_{1}(z)=Fz+\frac{a^{2}k}{z}+\frac{1}{\xi }{\left(z-ae^{i\psi }\right)}^{0.5+i\lambda }{\left(z-ae^{-i\psi }\right)}^{0.5-i\lambda }[(F-0.5)-\frac{D}{a^{2}z}]$$
(38)$$N_{2}=aFe^{-i\psi }+ake^{i\psi },\;N_{3}=Fae^{i\psi }+e^{-i\psi }ak$$
(39)$$M(z)=F-\frac{a^{2}k}{z^{2}}-[(F-0.5)z+H+\frac{C}{z}+\frac{D}{z^{2}}]\chi (z)$$
(40)$$F=\frac{1-(\cos\psi +2\lambda \sin\psi )\exp[2\lambda (\pi -\psi )]+(1-k)(1+4\lambda^{2})\sin^{2}\psi }{\frac{4}{k}-2-2(\cos\psi +2\lambda \sin\psi )\exp[2\lambda (\pi -\psi )]}$$
(41)H=a(cosψ+2λsinψ)(0.5−F)
(42)$$C=(k-1)(\cos\psi -2\lambda \sin\psi )a^{2}\exp[2\lambda (\psi -\pi )]$$
(43)$$D=(1-k)a^{3}\exp[2\lambda (\psi -\pi )]$$
(44)$$\chi (z)={\left(z-ae^{i\psi }\right)}^{-0.5+i\lambda }{\left(z-ae^{-i\psi }\right)}^{-0.5-i\lambda }$$
(45)$$k=\frac{G^{m}\left(1+\kappa_{2}\right)}{\left(1+\xi \right)(G^{m}+\kappa_{1}G_{23}^{f})},\;\lambda =-(\ln\xi )/(2\pi ),\;\xi =(G_{23}^{f}+\kappa_{2}G^{m})/(G^{m}+\kappa_{1}G_{23}^{f})$$
(46)$$\kappa_{1}=3-4v^{m},\kappa_{2}=\frac{3-\nu_{23}^{f}-4\nu_{12}^{f}\nu_{21}^{f}}{1+\nu_{23}^{f}},$$

In the above, *z* = *x*_2_ + *ix*_3_ and *i* =$\sqrt{-1}$; *ψ* is the half central angle corresponding to a steady state interface crack, which is determined through the following equations [8]:(47)ReG0−1k−2(1−k)kexp(iϕ)exp[2λ(ψ−π)]R(eiϕ)ϕ=ψ−γ=0
(48)$$R(\exp(i\phi ))={\left[\exp(i(\phi ))-e^{i\psi }\right]}^{0.5+i\lambda }{\left[\exp(i(\phi ))-e^{-i\psi }\right]}^{0.5-i\lambda }\exp(-i(\phi ))$$
(49)$$G_{0}=\frac{1-(\cos\psi +2\lambda \sin\psi )\exp[2\lambda (\pi -\psi )]+(1-k)(1+4\lambda^{2})\sin^{2}\psi }{2-k-k(\cos\psi +2\lambda \sin\psi )\exp[2\lambda (\pi -\psi )]}$$
(50)$$\gamma =\left\{\begin{array}{l} \frac{2\lambda (J_{1}^{2}+J_{2}^{2})}{J_{1}^{2}+J_{2}^{2}-2J_{2}J_{3}}{,}_{}if_{}\xi <1\\ -\frac{2\lambda (J_{1}^{2}+J_{2}^{2})}{J_{1}^{2}+J_{2}^{2}-2J_{2}J_{3}}{,}_{}if_{}\xi >1 \end{array}\right.$$
(51)$$J_{1}=kG_{0}-1-2(1-k)\xi \exp(2\lambda \psi )\cos(\psi )$$
(52)$$J_{2}=2(1-k)\xi \exp(2\lambda \psi )\sin(\psi )$$
(53)$$J_{3}=2(1-k)\xi \exp(2\lambda \psi )[J_{1}\cos(\psi )-J_{2}\sin(\psi )]/J_{2}$$

When *ξ* = 1, the crack angle *ψ* is indeterminate and the crack is named a singular crack. However, adjusting a fiber or matrix property parameter slightly, one can always attain *ξ* ≠ 1, since a deviation exists in measuring the parameter.

### 3.5. Matrix Longitudinal SCFs [10,11]

Only in a short fiber or particle composite (Figure 3b), can a longitudinal SCF of the matrix exist. No such SCF occurs in a continuous fiber composite, since the resulting matrix point-wise stress field is uniform [3]. Different from the derivations for all of the other matrix SCFs which are essentially based on the RVE geometry of Figure 3a, the RVE of Figure 3b used to derive the longitudinal SCFs of the matrix in a short fiber composite must be separated into three segments schematically shown in Figure 5, so that a longitudinal SCF of the matrix in a short fiber composite can become that in a continuous one when the fiber aspect ratio tends to infinity or when the fiber length ratio equals to 1. According to the condition that the matrix true stress resultant in the longitudinal direction in Figure 5a should equal to that in Figure 5b plus those in Figure 5c, one obtains (noting that the cross-sectional area in each figure is the same)
(54)$$2LK_{22}^{t}{\left(\sigma_{11}^{m}\right)}_{BM}^{L}=2(l-h){\left(\sigma_{11}^{m}\right)}_{BM}^{c}+2(L-l+h)K_{22}^{t,I}{\left(\sigma_{11}^{m}\right)}_{BM}^{h},$$
where
(55)$${\left(\sigma_{11}^{m}\right)}_{BM}^{L}=\frac{a_{11}^{L}\sigma_{11}}{V_{f}^{}+(1-V_{f}^{})a_{11}^{L}},$$
(56)$${\left(\sigma_{11}^{m}\right)}_{BM}^{c}=\frac{a_{11}^{c}\sigma_{11}}{V_{f}^{I}+(1-V_{f}^{I})a_{11}^{c}},$$
(57)$${\left(\sigma_{11}^{m}\right)}_{BM}^{h}=\frac{a_{11}^{h}\sigma_{11}}{V_{f}^{II}+(1-V_{f}^{II})a_{11}^{h}},$$
(58)$$V_{f}=(a/b{)}^{2}(l/L),\;V_{f}^{I}=(a/b{)}^{2}=V_{f}(L/l),\;V_{f}^{II}=(a/b{)}^{2}(h/(L-l+h))$$
(59)$$a_{11}^{c}=E^{m}/E_{11}^{f},$$
(60)$$a_{11}^{L}=\frac{V_{f}E^{m}}{(1-V_{f})E_{11}^{f}}\frac{\sigma_{11}L-E_{11}^{f}[\varepsilon_{L}^{1}l+\varepsilon_{L}^{2}(L-l)]}{E_{}^{m}[\varepsilon_{L}^{1}l+\varepsilon_{L}^{2}(L-l)]-\sigma_{11}L},$$
(61)$$a_{11}^{h}=\frac{V_{f}^{II}E^{m}}{(1-V_{f}^{II})E_{11}^{f}}\frac{\sigma_{11}L^{\prime }-E_{11}^{f}[\varepsilon_{L^{\prime }}^{1}h+\varepsilon_{L^{\prime }}^{2}(L^{\prime }-h)]}{E_{}^{m}[\varepsilon_{L^{\prime }}^{1}h+\varepsilon_{L^{\prime }}^{2}(L^{\prime }-h)]-\sigma_{11}L^{\prime }},$$
(62)$$L^{\prime }=L-l+h.$$

It is noted that Equation (61) is obtained from the second part of Equation (12), i.e., Equation (60), with *L* and *l* in Figure 3b and in Equations (16) replaced by *L*’ and *h*, respectively, in which *L*’ is given by Equation (62). The longitudinal tensile and compressive SCFs of the matrix in the short fiber composite are expressed below.
(63)$$K_{11}^{t}=\left\{\begin{array}{l} K_{11}^{t,I}{,}_{}if_{}l\leq h\\ \frac{(L-l+h){\left(\sigma_{11}^{m}\right)}_{BM}^{h}}{L{\left(\sigma_{11}^{m}\right)}_{BM}^{L}}K_{11}^{t,I}+\frac{l-h}{L}\frac{{\left(\sigma_{11}^{m}\right)}_{BM}^{c}}{{\left(\sigma_{11}^{m}\right)}_{BM}^{L}}{,}_{}if_{}l>h \end{array}\right.$$
(64)$$K_{11}^{c}=\left\{\begin{array}{l} K_{11}^{c,I}{,}_{}if_{}l\leq h\\ \frac{(L-l+h){\left(\sigma_{11}^{m}\right)}_{BM}^{h}}{L{\left(\sigma_{11}^{m}\right)}_{BM}^{L}}K_{11}^{c,I}+\frac{l-h}{L}\frac{{\left(\sigma_{11}^{m}\right)}_{BM}^{c}}{{\left(\sigma_{11}^{m}\right)}_{BM}^{L}}{,}_{}if_{}l>h \end{array}\right.$$
(65)$$K_{11}^{t,I}=\frac{(1-V_{f})E^{m}}{\sigma_{11}-\varepsilon_{L}^{1}E_{11}^{f}V_{f}}\left\{\frac{\sigma_{11}}{E^{m}}-\frac{D_{II}\sinh[n_{II}(L-l)][J_{0}(n_{II}a)-\frac{J_{1}(n_{II}b)}{Y_{1}(n_{II}b)}Y_{0}(n_{II}a)]}{\left(1+\nu^{m})(L-l\right)}\right\}$$
(66)$$K_{11}^{c,I}=\frac{1-V_{f}}{\sigma_{11}-\varepsilon_{L}^{1}E_{11}^{f}V_{f}}\left[\sigma_{11}-\frac{E^{m}}{2(1+\nu^{m})}\frac{D_{II}}{a}\sum_{k=1}^{N}\lambda_{k}f\left(\frac{b+a}{2}+\frac{b-a}{2}t_{k}\right)\right]$$
(67)$$f\left(r\right)=\cosh\left[n_{II}(r-a)\cot\phi \right]\left[J_{0}\left(n_{II}r\right)-\frac{J_{1}\left(n_{II}b\right)}{Y_{1}\left(n_{II}b\right)}Y_{0}\left(n_{II}r\right)\right]$$
(68)$$\varphi =\frac{\pi }{4}+\frac{1}{2}\arcsin\frac{\sigma_{u,c}^{m}-\sigma_{u,t}^{m}}{2\sigma_{u,c}^{m}}$$
(69)$$\frac{D_{II}}{a}=\frac{\sigma_{FD}-\sigma_{zz,II}^{f,0}}{n_{II}a\cosh[n_{II}(l-L)]\omega_{f,II}}$$
(70)$$\sigma_{FD}=\frac{\sigma_{zz,I}^{f,0}\omega_{I}\tanh(n_{I}l)+\sigma_{zz,II}^{f,0}\omega_{II}\tanh[n_{II}(L-l)]}{\omega_{I}\tanh(n_{I}l)+\omega_{II}\tanh[n_{II}(L-l)]}$$
(71)$$\sigma_{zz,I}^{f,0}=E_{11}^{f}\varepsilon_{L}^{1}+\frac{2\nu_{12}^{f}E^{m}}{\left(1+\nu^{m})(1-2\nu^{m}\right)}\left[\varepsilon_{L}^{1}\nu^{m}+C_{1}-(1-2\nu^{m})\frac{C_{2}}{a^{2}}\right]$$
(72)$$\sigma_{zz,II}^{f,0}=E^{m}\varepsilon_{L}^{2}+\frac{2\nu^{m}E^{m}}{\left(1+\nu^{m})(1-2\nu^{m}\right)}\left[\varepsilon_{L}^{2}\nu^{m}+\frac{(1+\nu^{m})(1-2\nu^{m})\sigma_{11}-(1-\nu^{m})\varepsilon_{L}^{2}E^{m}}{2\nu^{m}E^{m}}\right]$$
(73)$$\omega_{I}=\frac{\omega_{rz,I}}{\omega_{f,I}},\;\omega_{II}=\frac{\omega_{rz,II}}{\omega_{f,II}},\;\omega_{f,I}=-\frac{2}{n_{I}a}\omega_{rz,I},\;\omega_{f,II}=-\frac{2}{n_{II}a}\omega_{rz,II}$$
$$\omega_{rz,I}=\frac{E^{m}}{1+\nu^{m}}\left\{n_{I}a\left[J_{0}(n_{I}a)-\frac{J_{1}(n_{I}b)}{Y_{1}(n_{I}b)}Y_{0}(n_{I}a)\right]+\right.\;B_{53}J_{1}(n_{I}a)+B_{63}Y_{1}(n_{I}a)$$
(74)$$\left.+2(1-\nu^{m})\left[J_{1}(n_{I}a)-\frac{J_{1}(n_{I}b)}{Y_{1}(n_{I}b)}Y_{1}(n_{I}a)\right]\right\}$$
(75)$$\omega_{rz,II}=\frac{E^{m}}{1+\nu^{m}}\left[J_{1}(n_{II}a)-\frac{J_{1}(n_{II}b)}{Y_{1}(n_{II}b)}Y_{1}(n_{II}a)\right]$$
(76)$$\left[\begin{array}{l} D_{11}D_{12}\\ D_{21}D_{22} \end{array}\right]\left\{\begin{array}{l} B_{5}{}_{3}\\ B_{6}{}_{3} \end{array}\right\}=\left\{\begin{array}{l} M_{1}\\ M_{2} \end{array}\right\}$$
(77)$$D_{11}=\left(\frac{2\nu_{12}^{f}E^{m}}{1+\nu^{m}}-E_{11}^{f}\right)n_{I}aJ_{0}(n_{I}a)+\frac{2(1-\nu_{12}^{f})E^{m}}{1+\nu^{m}}J_{1}(n_{I}a)$$
(78)$$D_{12}=\left(\frac{2\nu_{12}^{f}E^{m}}{1+\nu^{m}}-E_{11}^{f}\right)n_{I}aY_{0}(n_{I}a)+\frac{2(1-\nu_{12}^{f})E^{m}}{1+\nu^{m}}Y_{1}(n_{I}a)$$
$$D_{21}=\frac{E_{11}^{f}n_{I}aJ_{0}(n_{I}a)}{\left(1+\nu^{m})[E_{11}^{f}-E_{22}^{f}{\left(\nu_{12}^{f}\right)}^{2}\right]}[\nu_{12}^{f}E^{m}(1+\nu_{23}^{f})-(1+\nu^{m})E_{11}^{f}]$$
(79)$$+\frac{E_{11}^{f}J_{1}(n_{I}a)\nu_{12}^{f}}{\left(1+\nu^{m})[E_{11}^{f}-E_{22}^{f}{\left(\nu_{12}^{f}\right)}^{2}\right]}[(1+\nu^{m})E_{22}^{f}-E^{m}(1+\nu_{23}^{f})]+\frac{2E^{m}J_{1}(n_{I}a)}{1+\nu^{m}}$$
$$D_{22}=\frac{E_{11}^{f}n_{I}aY_{0}(n_{I}a)}{\left(1+\nu^{m})[E_{11}^{f}-E_{22}^{f}{\left(\nu_{12}^{f}\right)}^{2}\right]}[\nu_{12}^{f}E^{m}(1+\nu_{23}^{f})-(1+\nu^{m})E_{11}^{f}]$$
(80)$$+\frac{E_{11}^{f}Y_{1}(n_{I}a)\nu_{12}^{f}}{\left(1+\nu^{m})[E_{11}^{f}-E_{22}^{f}{\left(\nu_{12}^{f}\right)}^{2}\right]}[(1+\nu^{m})E_{22}^{f}-E^{m}(1+\nu_{23}^{f})]+\frac{2E^{m}Y_{1}(n_{I}a)}{1+\nu^{m}}$$
$$M_{1}=2n_{I}a\left\{\frac{E^{m}[(1-2\nu^{m})\nu_{12}^{f}-1]}{1+\nu^{m}}+2(1-\nu^{m})E_{11}^{f}\right\}\left[J_{0}(n_{I}a)-\frac{J_{1}(n_{I}b)}{Y_{1}(n_{I}b)}Y_{0}(n_{I}a)\right]$$
(81)$$-\left\{\frac{2E^{m}[2(1-\nu^{m})+{\left(n_{I}a\right)}^{2}\nu_{12}^{f}]}{1+\nu^{m}}+{\left(n_{I}a\right)}^{2}E_{11}^{f}\right\}\left[J_{1}(n_{I}a)-\frac{J_{1}(n_{I}b)}{Y_{1}(n_{I}b)}Y_{1}(n_{I}a)\right]$$
$$M_{2}=\frac{n_{I}a}{1+\nu^{m}}\left\{m_{1}\left[J_{0}(n_{I}a)-\frac{J_{1}(n_{I}b)}{Y_{1}(n_{I}b)}Y_{0}(n_{I}a)\right]\right.$$
(82)$$\left.+m_{2}\left[J_{1}(n_{I}a)-\frac{J_{1}(n_{I}b)}{Y_{1}(n_{I}b)}Y_{1}(n_{I}a)\right]\right\}$$
(83)$$m_{1}=\frac{E^{m}\nu_{12}^{f}(1+\nu_{23}^{f})(1-2\nu^{m})-4E_{11}^{f}[1-{\left(\nu^{m}\right)}^{2}]+\nu_{12}^{f}E_{22}^{f}(1+\nu^{m})}{1-E_{22}^{f}{\left(\nu_{12}^{f}\right)}^{2}/E_{11}^{f}}-2E^{m}$$
(84)$$m_{2}=\frac{n_{I}a[E^{m}\nu_{12}^{f}(1+\nu_{23}^{f})-E_{11}^{f}(1+\nu^{m})]}{1-E_{22}^{f}{\left(\nu_{12}^{f}\right)}^{2}/E_{11}^{f}}-\frac{4E^{m}(1-\nu^{m})}{n_{I}a}$$
(85)$$B_{53}J_{1}(n_{I}b)Y_{1}(n_{I}b)+B_{63}Y_{1}^{2}(n_{I}b)-n_{I}b[Y_{0}(n_{I}b)J_{1}(n_{I}b)-J_{0}(n_{I}b)Y_{1}(n_{I}b)]=0$$
(86)$$\left[n_{II}aJ_{0}(n_{II}a)-2J_{1}(n_{II}a)\right]Y_{1}(n_{II}b)-\left[n_{II}aY_{0}(n_{II}a)-2Y_{1}(n_{II}a)\right]J_{1}(n_{II}b)=0$$
(87)$$[\sigma_{FD}-\sigma_{zz,I}^{f,0}]\frac{\cosh[n_{I}(l-h)]}{\cosh(n_{I}l)}+\sigma_{zz,I}^{f,0}=0.99{\bar{\sigma }}_{zz}^{f}(0),\mathrm{if}\;l/a\leq 100$$
(88)$$[\sigma_{FD}-\sigma_{zz,I}^{f,0}]\frac{\cosh[n_{I}a(100-h/a)]}{\cosh(100n_{I}a)}+\sigma_{zz,I}^{f,0}=0.99{\bar{\sigma }}_{zz}^{f}(0),\mathrm{if}\;l/a>100$$
(89)$${\bar{\sigma }}_{zz}^{f}(0)=\left\{\begin{array}{l} {}[\sigma_{FD}-\sigma_{zz,I}^{f,0}]/\cosh(n_{I}l)+\sigma_{zz,I}^{f,0}{,}_{}if_{}l/a\leq 100\\ \approx \sigma_{zz,I}^{f,0}{,}_{}if_{}l/a>100 \end{array}\right.$$

In the above, *J*_0_, *J*_1_, *Y*_0_, and *Y*_1_ are, respectively, the zero- and one-order Bessel functions of the first and second classes, and *a* and *b* are the radii of the fiber and matrix cylinders in the RVE (Figure 3b), respectively. *C*_1_, *C*_2_, $\varepsilon_{L}^{1}$, and $\varepsilon_{L}^{2}$ are the coefficients solved from Equations (16) to (20), and *g* is defined in Equation (17). In Equation (66), *N* is the number of a Gaussian integration, and *t_k_* and *λ_k_* are the integration points and weighing coefficients, respectively. In general, taking *N* = 5 can attain an enough high accuracy.

### 3.6. Numerical Examples

The matrix SCF formulae under a transverse tension after interface debonding and under a longitudinal load in a short fiber composite are extremely complicated, as the elasticity solutions to the matrix stress fields themselves are very complicated [18,26,27]. After line integration, the complication level is elevated one order higher. Fortunately, this author implemented all of the SCF formulae, Equations (27)–(53) and Equations (63)–(89), into an Excel table-based program to calculate them. For any question in using the program, the reader can write to the author. The SCFs of the matrices in the nine independent material systems used in three world-wide failure exercises (WWFEs) [28,29,30] are summarized in Table 1, whereas the input data for the calculations taken from [28,29,30] are listed in Table 2.

Simply speaking, a relative difference between the predicted strengths of a composite made from the nine material systems based on the homogenized and the true stresses will be at least 34% and at most 669%, as indicated in Table 1.

### 3.7. True Stresses

Let $\{d\sigma_{i}^{m}\}={\left\{d\sigma_{11}^{m},d\sigma_{22}^{m},d\sigma_{33}^{m},d\sigma_{23}^{m},d\sigma_{13}^{m},d\sigma_{12}^{m}\right\}}^{T}$ be the homogenized stress increments of the matrix due to an arbitrary load increments {*dσ_j_*} applied on the RVE of either Figure 3a or Figure 3b. It is evaluated from Equation (4) by replacing {*σ_j_*} with {*dσ_j_*} on its right hand side. The overall true stresses of the matrix are updated through
(90)$${\left\{{\bar{\sigma }}_{i}^{m}\right\}}_{n}={\left\{{\bar{\sigma }}_{11}^{m},{\bar{\sigma }}_{22}^{m},{\bar{\sigma }}_{33}^{m},{\bar{\sigma }}_{23}^{m},{\bar{\sigma }}_{13}^{m},{\bar{\sigma }}_{12}^{m}\right\}}_{n}={\left\{{\bar{\sigma }}_{i}^{m}\right\}}_{n-1}+\{d{\bar{\sigma }}_{i}^{m}\},n=1,2,\ldots$$
(91)$$\{d{\bar{\sigma }}_{i}^{m}\}={\left\{K_{11}d\sigma_{11}^{m},K_{22}d\sigma_{22}^{m},K_{33}d\sigma_{33}^{m},K_{23}d\sigma_{23}^{m},{\bar{K}}_{12}d\sigma_{13}^{m},{\bar{K}}_{12}d\sigma_{12}^{m}\right\}}_{\;}$$
where
(92)$$K_{11}=\left\{\begin{array}{l} 1,for^{}\;continuous^{}\;fiber^{}\;composite\\ K_{11}^{t},for^{}\;other^{}\;composite_{}\;with_{}\;d\sigma_{11}^{m}\geq 0\\ K_{11}^{c},for^{}\;other^{}\;composite_{}\;with_{}\;d\sigma_{11}^{m}<0 \end{array}\right.$$
(93)$$K_{22}=\left\{\begin{array}{l} K_{22}^{t},if^{}\;d\sigma_{22}^{m}>0^{}\;with^{}\;perfect^{}\;\mathrm{int}erface\\ {\hat{K}}_{22}^{t},if^{}\;d\sigma_{22}^{m}>0^{}\;with^{}\;debonded^{}\;\mathrm{int}erface\\ K_{22}^{c},if^{}d\sigma_{22}^{m}<0 \end{array}\right.$$
(94)$$K_{33}=\left\{\begin{array}{l} K_{22}^{t},if^{}\;d\sigma_{33}^{m}>0^{}\;with^{}\;perfect^{}\;\mathrm{int}erface\\ {\hat{K}}_{22}^{t},if^{}\;d\sigma_{33}^{m}>0^{}\;with^{}\;debonded^{}\;\mathrm{int}erface\\ K_{22}^{c},if^{}\;d\sigma_{33}^{m}<0 \end{array}\right.$$
(95)$${\bar{K}}_{12}=\left\{\begin{array}{l} K_{12},\mathrm{with}\;perfect^{}\;\mathrm{int}erface\\ {\hat{K}}_{12},\mathrm{with}\;debonded^{}\;\mathrm{int}erface \end{array}\right.$$

The fiber true stresses are updated from:(96)$${\left\{{\bar{\sigma }}_{i}^{f}\right\}}_{n}={\left\{{\bar{\sigma }}_{i}^{f}\right\}}_{n-1}+\{d\sigma_{j}^{f}\},n=1,2,\ldots$$

It is noted that the internal stresses occurred in Figure 4 must be the internal true stresses.

### 3.8. Roles of the True Stresses

Eshelby [1] found that if a solid embedded in an unbounded matrix is an ellipsoid, the stress field in it is uniform no matter what kind of load is applied to the matrix. A continuous fiber in a composite can be regarded as an infinitely long ellipsoid, whereas a short fiber or particle can be considered as an ellipsoid of a finite length. Thus, the Eshelby’s finding implies that one can estimate any failure of the fiber in a composite by directly comparing its stresses against the strengths of the monolithic fiber determined independently, since the stress fields in obtaining the fiber strengths are uniform. This also explains why a fiber controlled composite failure can be estimated reasonably by anybody in general.

However, the stress field in the matrix outside the fiber is not uniform. Any failure of the matrix in a composite cannot be assessed by directly comparing its stresses against the strengths of the monolithic matrix measured independently. The reason is in that the matrix strengths are measured in uniform stress fields, whereas the matrix stresses in the composite are not uniform. The two kinds of quantities are not comparable. To estimate any matrix-induced composite failure, one must firstly convert the non-uniform stresses of the matrix into equivalent, uniform ones. Otherwise, one cannot obtain the critical stress or deformation data of the matrix to detect any of the matrix-dominated composite properties outside a linear elastic range, such as a plastic behavior, failure, strength, etc. Simply speaking, the true stresses of the matrix determined through Equation (90) stand for the uniform values.

## 4. Selected Applications

### 4.1. Uniaxial Strengths of UD Composites

First, substitute the material property data of Table 2 into Equations (6)–(11) to calculate the homogenized internal stresses of the composite subjected to uniaxial load one by one, and let the major internal stress attain either the fiber or matrix strength of Table 2. The resulting external load is taken as the composite strength. Except for the longitudinal strength, which is assumed to be caused by a fiber failure, all of the other uniaxial strengths are resulted from the matrix failures. They are named homogenized stresses-based predictions. The averaged errors between the predictions and the experiments are summarized in Table 3. Second, after converting the homogenized stresses into the true values and assuming that the interface bonding for each of the composites is perfect up to failure, similar predictions can be made. Again, the averaged errors for the true stresses-based predictions are listed in the table as well.

From Table 3, the following conclusions can be made: (1) Only a longitudinal strength is independent of a true stress effect. (2) The overall averaged error of the predictions based on the homogenized stresses is 3.92 times bigger than that upon the true stresses. If the longitudinal strengths are excluded, the difference becomes 5.12 times. (3) Using the true stresses, the averaged prediction errors bigger than 20% are for the composite strengths under longitudinal compression, transverse tension, and transverse compression. The error for the longitudinal compressive strength comes from an initial fiber misalignment; the error for the transverse compressive strength is due to the fact that the outward normal of the resulting failure surface is always perpendicular to the fiber axis; and the error for the transverse tensile strength is attributed to the interface debonding. How to reduce these errors will be highlighted subsequently. (4) The predictions for the in-plane and transverse shear strengths are high enough, indicating that an interface debonding has a negligible effect on the shear load sustaining ability of the composite.

In fact, if an interface debonding is taken into account, the in-plane shear strength formula is changed to
(97)$$\sigma_{12}^{u}=(V_{f}+V_{m}a_{66})[{\hat{\sigma }}_{e}^{m}/\sqrt{3}+(\sigma_{u,s}^{m}-K_{12}{\hat{\sigma }}_{e}^{m}/\sqrt{3})/{\hat{K}}_{12}^{}]/a_{66}$$
where ${\hat{\sigma }}_{e}^{m}$ is determined by a subsequent Equation (101). Having done so, the overall error for the in-plane shear strengths is changed from 13.1% (Table 3) to 11.1%.

Any experiment has a deviation. The measurement errors exist for not only the constituent properties and the fiber volume content, but also the composite property data. All of them gathered together generate a cumulative effect. Thus, an expected overall error between measured and predicted elastic moduli of the composites using independent constituent properties and fiber geometric parameters can only be 10%. Indeed, this error is 10.4% [16] if the data of Table 2 are substituted into Equations (21)–(25). An expectation for an even higher accuracy is unrealistic in general. Similarly, an expected overall error for strength prediction of a composite can be as small as 20% if only the independent constituent properties and the fiber content are available, since the prediction needs more input data and a larger cumulative error is involved. A higher prediction accuracy depends on the input of some composite ultimate data. Table 3 indicates that a failure and strength prediction for a composite in terms of the bridging model and the matrix true stress theory is near to the expectation.

### 4.2. Interface Debonding

It is known in the composite community that most composite failures are initiated from fiber and matrix interface debonding. However, given an arbitrary load, when does the interface debonding occur? Both the existing studies [31,32,33] and Table 3 indicate that the interface debonding only has an effect on the transverse tensile load sustaining ability of the composite.

Let the composite under a transverse tension *σ*_22_ attain an interface debonding at ${\hat{\sigma }}_{22}^{}$. Due to a Poisson’s retardant force in the direction perpendicular to the loading, as shown in Figure 6, the interface debonding soon reaches stable, and a load variation from the initial to the stable debonding stages is negligible. One can assume that the interface before the critical load ${\hat{\sigma }}_{22}^{}$ is in perfect bonding, and after it, in stable debonding. The transverse failure condition reads
(98)$$K_{22}^{t}{\hat{\sigma }}_{22}^{m}+{\hat{K}}_{22}^{t}(\sigma_{22}^{m,Y}-{\hat{\sigma }}_{22}^{m})=\sigma_{u,t}^{m}$$

In Equation (98), ${\hat{\sigma }}_{22}^{m}$ and $\sigma_{22}^{m,Y}$ are the matrix homogenized stresses by replacing *σ*_22_ in Equation (9) with ${\hat{\sigma }}_{22}^{}$ and *Y*, respectively, where *Y* is the transverse tensile strength of the composite. From Equation (98), one obtains the critical transverse load as
(99)$${\hat{\sigma }}_{22}^{}=\frac{{\hat{K}}_{22}^{t}Y}{{\hat{K}}_{22}^{t}-K_{22}^{t}}-\frac{(V_{f}+0.3V_{m})E_{22}^{f}+0.7V_{m}E^{m}}{\left(0.3E_{22}^{f}+0.7E^{m})({\hat{K}}_{22}^{t}-K_{22}^{t}\right)}\sigma_{u,t}^{m}$$

A necessary and sufficient condition for an interface debonding in any composite subjected to an arbitrary load is given by
(100)$${\bar{\sigma }}_{e}^{m}\geq {\hat{\sigma }}_{e}^{m}\mathrm{and}{\bar{\sigma }}_{m}^{1}>0$$
(101)$${\hat{\sigma }}_{e}^{m}=\sqrt{{\left({\hat{\sigma }}_{11}^{m}\right)}^{2}+{\left(K_{22}^{t}{\hat{\sigma }}_{22}^{m}\right)}^{2}-K_{22}^{t}{\hat{\sigma }}_{11}^{m}{\hat{\sigma }}_{22}^{m}}$$

In the above equation, ${\hat{\sigma }}_{11}^{m}$ is calculated from Equation (7) with *σ*_22_ replaced by ${\hat{\sigma }}_{22}^{}$ and *σ*_11_ = *σ*_33_ = 0; ${\bar{\sigma }}_{e}^{m}$ and ${\bar{\sigma }}_{m}^{1}$ are the Mises and the first principal true stresses calculated from Equation (90), respectively. Therefore, in addition to the fiber and matrix properties, one only needs to know the transverse tensile strength *Y* of the composite.

### 4.3. Interface Slippage

An advanced, e.g., carbon fiber-reinforced composite is generally considered to be brittle, since its load-deflection curves in both the longitudinal and transverse directions are essentially linear up to failure. However, significant non-linear deformation can occur in a composite, especially when it is subjected to an in-plane shear or an off-axial load [34]. The in-plane shear nonlinear deformation of the composite can be even bigger than the elastic–plastic deformation of the pure matrix. Shear deformation of the composite is one of the most difficult deformations to be analyzed [35]. We found [15] that a relative slippage displacement between debonded fiber and matrix interfaces is the key reason for the composite under an in-plane shear to generate a large deformation, as indicated in Figure 7. Before the interface debonding, a longitudinal plane denoted by the line FD in a transverse cross-section (Figure 7b) deforms as schematically shown in Figure 7a. After the debonding, the point D further moves to D’ (Figure 7a).

Under a transverse tension, an interface debonding will soon reach stable due to a Poisson’s internal retardant (Figure 6), resulting in hardly any effect on a transverse tensile stress–strain curve. In the contrast, no such retardant exists in the case of an in-plane shear loading. An interface debonding will propagate until there is a complete separation between the fiber and the matrix. Thus, as long as a fiber pulling out is seen, there must be a shear load component or a bi-axial transverse tension. Explicit formulae for an interface debonding-induced shear slippage displacement were derived in [15,20]. Its application to the prediction of shear deformation for the E-glass/MY750 UD composite [28] is shown in Figure 8. More details can refer to [20].

Therefore, a nonlinear constitutive relation of the composite is attributed to three contributions: (1) the matrix plasticity or nonlinear deformation, which is well known in the composite community, (2) the matrix true stress effect, which occurs in an instantaneous compliance tensor of the matrix (by any classical theory, a matrix instantaneous compliance tensor outside linear elastic range contains the current matrix stresses, which must be true values in the composite), and (3) a relative slippage displacement between debonded fiber and matrix interfaces.

### 4.4. Fiber Kinking or Splitting Failure

A particular phenomenon was already recognized in engineering. Namely, the fiber strength from, e.g., T300 to T1100 is elevated significantly, but the longitudinal compressive strength of the resulting composite almost keeps unchanged. Perhaps the most well known explanation for this was attributed to a matrix shear nonlinearity [36]. However, even for a very brittle matrix, the phenomenon still exists. Using the matrix true stress theory, the phenomenon can be displayed accurately.

Let $\theta_{c}^{f,0}$ be an initial fiber misalignment angle in the composite, which is almost inevitable to occur during a composite fabrication. A longitudinal compression in the misaligned coordinate system (Figure 3c) has a shear stress component, which will cause a misalignment angle increment $\theta_{c}^{f,1}$. Referring to Figure 9, external loads {*σ*_11_,*σ*_22_, *σ*_12_} generate three stress components in the local coordinate system, ($x_{1}^{I}$,$x_{2}^{I}$), as
(102)$$\sigma_{11}^{I}=\frac{\sigma_{11}^{}+\sigma_{22}^{}}{2}+\frac{\sigma_{11}^{}-\sigma_{22}^{}}{2}\cos(2\theta_{c}^{f})+\sigma_{12}^{}\sin(2\theta_{c}^{f})$$
(103)$$\sigma_{22}^{I}=\frac{\sigma_{11}^{}+\sigma_{22}^{}}{2}-\frac{\sigma_{11}^{}-\sigma_{22}^{}}{2}\cos(2\theta_{c}^{f})-\sigma_{12}^{}\sin(2\theta_{c}^{f})$$
(104)$$\sigma_{12}^{I}=-\frac{\sigma_{11}^{}-\sigma_{22}^{}}{2}\sin(2\theta_{c}^{f})+\sigma_{12}^{}\cos(2\theta_{c}^{f})$$
where $\theta_{c}^{f}$=$\theta_{c}^{f,0}$+$\theta_{c}^{f,1}$. Equations (102)–(104) are substituted into Equations (6)–(11) to calculate the homogenized stresses, and further into Equations (90) and (96) to obtain the true stresses. Letting the matrix attain a failure status, the misalignment angle increment $\theta_{c}^{f,1}$ is determined as [20]
(105)$$\theta_{c}^{f,1}=\frac{\sigma_{12}^{eq}(1-V_{f})\pi }{2G^{m}[\sqrt{V_{f}}\sqrt{1-V_{f}}+\sin^{-1}(\sqrt{V_{f}})]}$$
(106)$$\sigma_{12}^{eq}=\frac{V_{f}+V_{m}a_{66}}{K_{12}a_{66}}\sqrt{\frac{({\bar{\sigma }}_{11}^{m}+{\bar{\sigma }}_{22}^{m})c_{1}-1-2c_{1}e_{1}}{2c_{1}^{2}}}$$
(107)$$c_{1}=\frac{2\sigma_{u,c}^{m}}{4{\left(\sigma_{u,s}^{m}\right)}^{2}-{\left(\sigma_{u,c}^{m}\right)}^{2}},$$
(108)$$e_{1}=-\frac{{\left[4{\left(\sigma_{u,s}^{m}\right)}^{2}+{\left(\sigma_{u,c}^{m}\right)}^{2}\right]}^{2}}{8(\sigma_{u,c}^{m})[4{\left(\sigma_{u,s}^{m}\right)}^{2}-{\left(\sigma_{u,c}^{m}\right)}^{2}]}.$$

It is noted that the matrix true stresses in Equation (106), ${\bar{\sigma }}_{11}^{m}$ and ${\bar{\sigma }}_{22}^{m}$, should be expressed in the local system ($x_{1}^{I}$,$x_{2}^{I}$). If the corresponding load at the matrix failure is smaller than the fiber fracture load, the composite is said to assume a fiber kinking failure. Otherwise, the fiber fracture is attained. In the former case, a further increase in the fiber strength has essentially no contribution to the composite longitudinal strength. In the latter one, the longitudinal compressive strength of the composite reaches its maximum.

The initial fiber misalignment angle, $\theta_{c}^{f,0}$, is not easy to measure. Yurgartis reported that more than 80% of the initial misalignment angles were around 1^0^ [37]. Through a numerical test, we found that $\theta_{c}^{f,0}$=1.5^0^ resulted in the best prediction for the longitudinal compressive strengths of the nine composites in Table 1, with an overall correlation error from 25.1% in Table 3 (where $\theta_{c}^{f,0}$=0^0^) reduced to 16.1% [20].

A fiber splitting failure can be analyzed similarly [13]. The only difference is in that the misalignment increment $\theta_{c}^{f,1}$ is positive under a longitudinal compression, but negative under a longitudinal tension [13].

### 4.5. Matrix Compression-Induced Composite Failure

Except for a longitudinal compression on a composite, a failure of which can be caused by either a fiber fracture or a matrix shear failure, a matrix compression-dominated composite failure has two distinct characteristics (Figure 10): (1) the failure surface has an inclined angle with the loading direction [38,39]; and (2) the outward normal to the failure surface is generally perpendicular to the fiber axis.

In accordance with the above failure characteristics, a physics-based failure criterion was developed by the author et al. to detect a matrix compression-dominated composite failure [40,41]:(109)$$f(\sigma_{n}^{m},\tau_{n}^{m})=\tau_{n}^{m}+\mu \sigma_{n}^{m}=\sqrt{{\left(\tau_{ns}^{m}\right)}^{2}+{\left(\tau_{nt}^{m}\right)}^{2}}+\mu \sigma_{n}^{m}\geq \sigma_{u,s}^{m}/K_{n}$$
(110)$$K_{n}=\frac{\sqrt{{\left(K_{12}\tau_{ns}^{m}\right)}^{2}+{\left(K_{23}\tau_{nt}^{m}\right)}^{2}}}{\sqrt{{\left(\tau_{ns}^{m}\right)}^{2}+{\left(\tau_{nt}^{m}\right)}^{2}}}$$
(111)$$\mu =\left\{\begin{array}{l} 0.5,if_{}\;UD_{}\;composite_{}\;is_{}\;under_{}\;2D_{}\;compression\\ 0.2,if_{}\;UD_{}\;composite_{}\;is_{}\;under_{}\;3D_{}\;compression \end{array}\right.$$
(112)$$\left\{\begin{array}{l} \sigma_{n}^{m}(\theta )=\sigma_{22}^{m}\cos^{2}\theta +\sigma_{33}^{m}\sin^{2}\theta +2\sigma_{23}^{m}\cdot \sin\theta \cdot \cos\theta \\ \tau_{nt}^{m}(\theta )=(\sigma_{33}^{m}-\sigma_{22}^{m})\sin\theta \cos\theta +\sigma_{23}^{m}(\cos^{2}\theta -\sin^{2}\theta )\\ \tau_{ns}^{m}(\theta )=\sigma_{13}^{m}\sin\theta +\sigma_{12}^{m}\cos\theta \end{array}\right.$$
in which *θ* is the inclined angle between the failure surface (parallel to the fiber axis) and the *x*_3_ coordinate (in thickness direction of the UD composite) shown in Figure 11.

Using Equations (109)–(112), the predicted transverse compressive strengths of the nine composites in Table 1 have a much better correlation with the experiments, and the relative error of 23.2% in Table 3 is reduced to 13.8% [20,40].

### 4.6. Applications to Other Areas

The matrix true stress theory was also applied by many other researchers in their analysis for the failure and strength behaviors of composites under different load conditions. An incomplete list of the published papers by the other people using the matrix true stress theory is given in [42,43,44,45,46,47,48,49,50,51,52,53,54,55,56,57,58,59,60,61,62,63].

## 5. Conclusions

Eshelby’s work explained why a fiber-dominated composite failure can be estimated reasonably, since the uniform fiber stresses in any composite under arbitrary loads are comparable with the fiber strengths determined in uniform stress fields. The matrix stresses in the composite, however, are not uniform, and hence any matrix-dominated composite failure, as well as nonlinear behavior, cannot be assessed by comparing the non-uniform matrix stresses with its monolithic strengths or other critical parameters measured in uniform stress states. The two kinds of quantities of the matrix are not comparable. A conversion of the non-uniform matrix stresses into equivalent uniform ones must be carried out. The matrix true stresses stand for its equivalent uniform values, and the conversion is achieved through the author’s matrix true stress theory summarized in this paper. As most composite failures resulted from matrix failures, none of them can be properly estimated without the matrix true stresses. On the other hand, almost all of the mechanics of composite problems can be efficiently resolved based on the matrix true stress theory. Only some typical applications of the theory are highlighted in the paper. Analytical formulae for almost all of the true stress components were derived by the author, and are summarized in the paper. They were implemented into an Excel table-based program. If the reader has any question relevant to the Excel table-based program for calculating all of the matrix SCFs in a composite, he/she is welcome to write to the author at huangzm@tongji.edu.cn.

## Figures and Tables

**Figure 1 materials-16-00774-f001:**
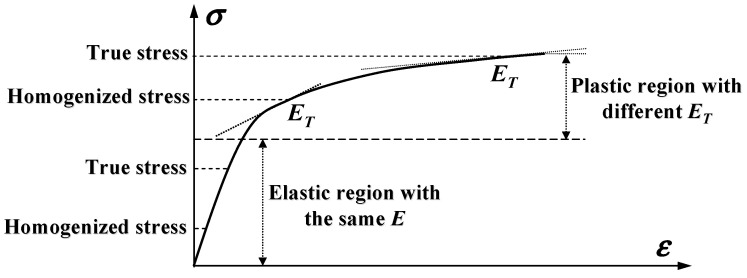
Schematic composite properties at different scales of stresses.

**Figure 2 materials-16-00774-f002:**
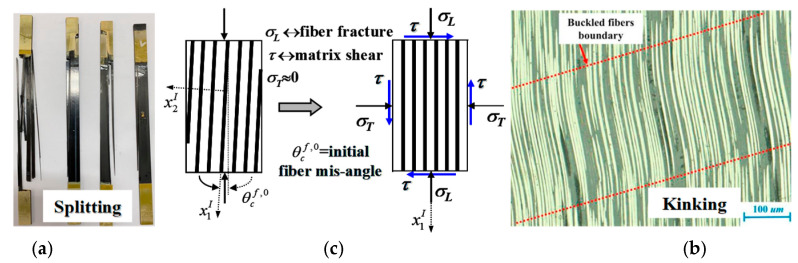
(**a**) Longitudinal tension-induced fiber splitting failure, (**b**) longitudinal compression-induced fiber kinking failure [12], (**c**) load decomposition in fiber misligned coordinate system.

**Figure 3 materials-16-00774-f003:**
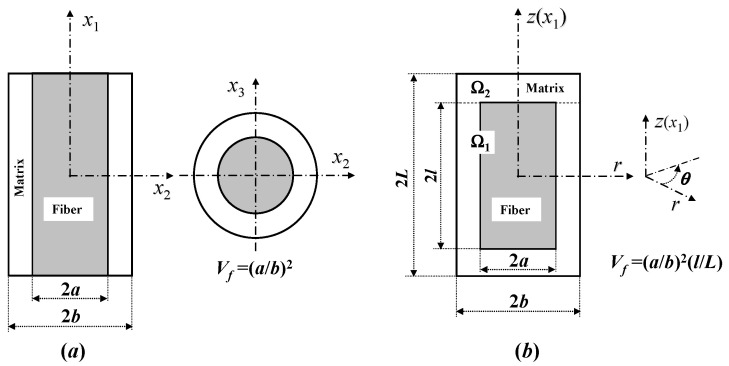
RVEs of (**a**) continuous fiber, and (**b**) short fiber reinforced composites.

**Figure 4 materials-16-00774-f004:**
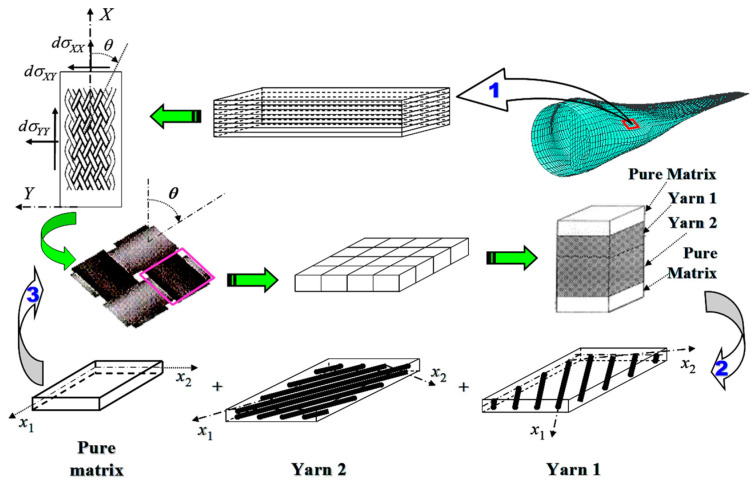
Analysing steps for any composite structure: 1. the structure is discretized into finite elements, and a laminate theory is applied to determine load share by each layer of the laminated element; 2. the layer is cut into a series of sheets each containing at most one straight fiber yarn segment, the bridging model is applied to determine the homogenized internal stresses and compliance tensor of each sheet, and the homogenized stresses are converted into true values as per the methods described subsequently; and 3. an assemblage from all of the sheets give the internal true stresses and compliance tensor of the layer.

**Figure 5 materials-16-00774-f005:**
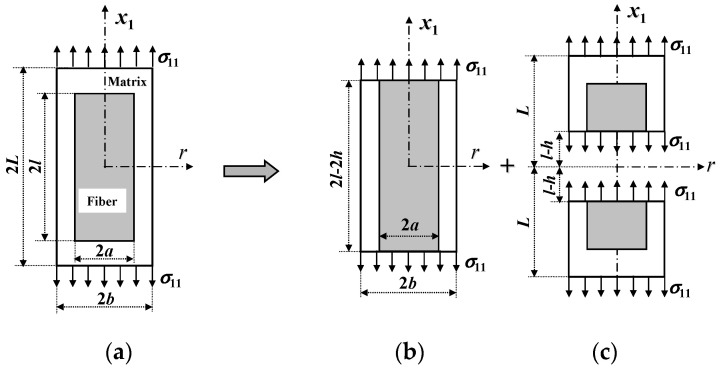
RVE of a short fiber composite (**a**) is separated into the RVE of a continuous fiber composite (**b**) plus two end segments of a short fiber composite with a small fiber aspect ratio (**c**).

**Figure 6 materials-16-00774-f006:**
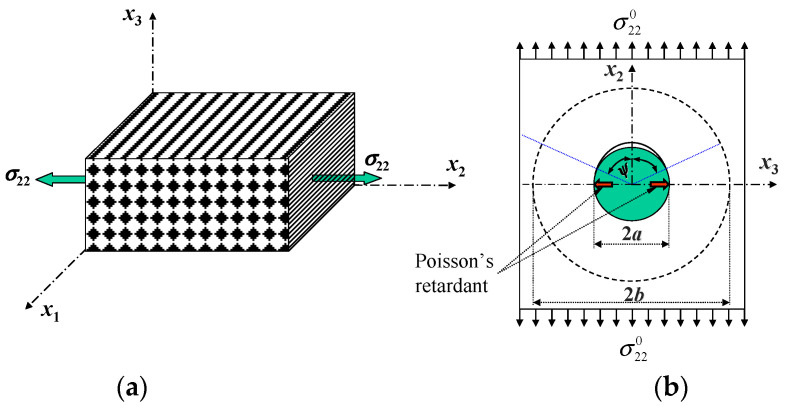
Schematic of (**a**) UD composite subjected to a transverse tension, and (**b**) stable interface debonding due to Poisson’s retardant.

**Figure 7 materials-16-00774-f007:**
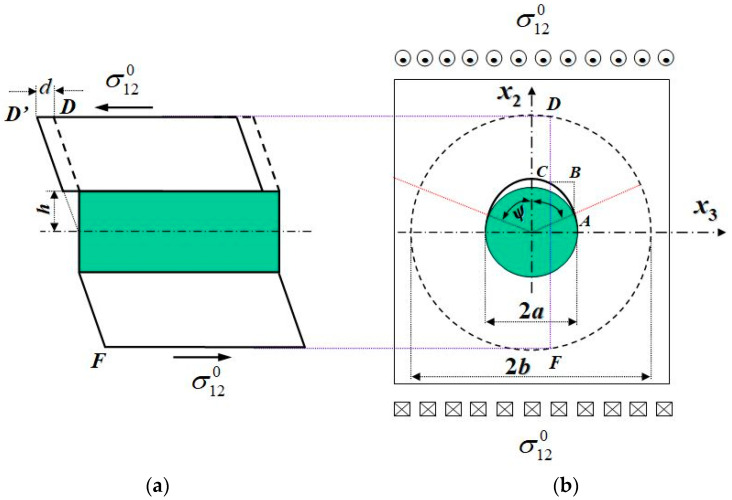
Schematic diagram for relative slippage of a RVE subjected to an in-plane shear shown in (**a**) a longitudinal cross-section, (**b**) a transverse cross-section.

**Figure 8 materials-16-00774-f008:**
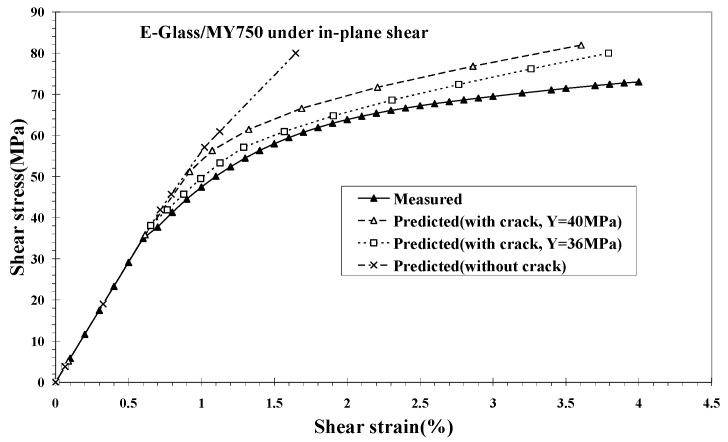
In-plane shear stress–strain curve of E-glass/MY750 up to failure.

**Figure 9 materials-16-00774-f009:**
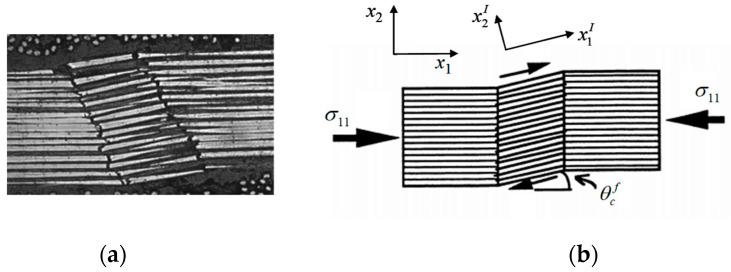
(**a**) Longitudinal compression-induced kink-band failure, (**b**) load analysis in kink band coordinate system.

**Figure 10 materials-16-00774-f010:**
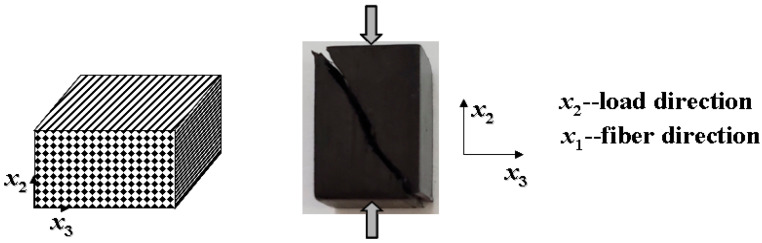
Failure surface of a UD composite under a compression.

**Figure 11 materials-16-00774-f011:**
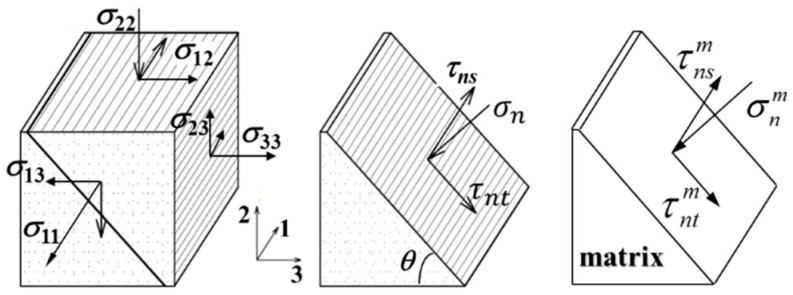
Schematic of matrix compression-induced composite failure surface.

**Table 1 materials-16-00774-t001:** Matrix SCFs with other relevant parameters of the nine composite systems used in WWFEs.

	E-Glass LY556	E-Glass MY750	AS4 3501-6	T300 BSL914C	IM7 8511-7	T300 PR319	AS Epoxy	S2-Glass Epoxy	G400-800 5260
*K* _12_	1.520	1.491	1.424	1.430	1.475	1.510	1.449	1.500	1.483
*K* _23_	3.020	2.936	1.337	2.421	2.034	2.167	1.999	2.982	2.469
$K_{22}^{t}$	3.339	3.253	2.098	2.143	2.327	3.123	2.339	3.317	2.464
$K_{22}^{c}$	2.249	2.181	1.469	1.570	1.761	2.035	1.743	2.172	1.732
${\hat{K}}_{12}^{}$	1.870	1.840	1.760	1.760	1.830	1.760	1.760	1.850	1.830
${\hat{K}}_{22}^{t}$	7.690	7.220	4.950	5.040	5.410	6.970	5.430	7.340	5.680
*ψ*	71.8^0^	71.9^0^	73.9^0^	73.9^0^	73.4^0^	72.0^0^	73.3^0^	71.8^0^	72.8^0^
*h*/*a*	0.609	0.666	0.563	0.574	0.648	0.673	0.591	0.680	0.627
$K_{11}^{t,I}$ *	3.660	3.574	3.733	3.877	4.486	4.188	3.953	3.688	4.024
$K_{11}^{t}$ *	1.598	1.675	1.516	1.588	1.965	1.844	1.636	1.726	1.720
$K_{11}^{c,I}$	3.510	3.419	3.550	3.685	4.271	3.995	3.759	3.528	3.833
$K_{11}^{c}$ *	1.504	1.577	1.427	1.496	1.860	1.757	1.545	1.631	1.630

* *l*/*a* = 2, *l/L* is determined from Equation (26) in which *c* = 0.03.

**Table 2 materials-16-00774-t002:** Constituent properties and other parameters of the nine composite systems used in WWFEs [28,29,30].

	E-Glass LY556	E-Glass MY750	AS4 3501-6	T300 BSL914C	IM7 8511-7	T300 PR319	AS Epoxy	S2-Glass Epoxy	G400-800 5260
$E_{11}^{f}$ (GPa)	80	74	225	230	276	230	231	87	290
$E_{22}^{f}$ (GPa)	80	74	15	15	19	15	15	87	19
$\nu_{12}^{f}$	0.2	0.2	0.2	0.2	0.2	0.2	0.2	0.2	0.2
$G_{12}^{f}$ (GPa)	33.33	30.8	15	15	27	15	15	36.3	27
$\nu_{23}^{f}$	0.2	0.2	0.07	0.07	0.36	0.07	0.07	0.2	0.357
$\sigma_{u,t}^{f}$ (MPa)	2150	2150	3350	2500	5180	2500	3500	2850	5860
$\sigma_{u,c}^{f}$ (MPa)	1450	1450	2500	2000	3200	2000	3000	2450	3200
*E^m^* (GPa)	3.35	3.35	4.2	4	4.08	0.95	3.2	3.2	3.45
*ν^m^*	0.35	0.35	0.34	0.35	0.38	0.35	0.35	0.35	0.35
$\sigma_{u,t}^{m}$ (MPa)	80	80	69	75	99	70	85	73	70
$\sigma_{u,c}^{m}$ (MPa)	120	120	250	150	130	130	120	120	130
$\sigma_{u,s}^{m}$ (MPa)	54	54	50	70	57	41	50	52	57
*Y* (MPa)	35	40	48	27	73	40	38	63	75
*V_f_*	0.62	0.6	0.6	0.6	0.6	0.6	0.6	0.6	0.6

**Table 3 materials-16-00774-t003:** Averaged errors between predictions and measurements for uniaxial strengths of the nine UD composites used in WWFEs.

	Longitudinal Tensile strengths	Longitudinal Compressive Strengths	Transverse Tensile Strengths	Transverse Compressive Strengths	In-Plane Shear Strengths	Transverse Shear Strengths	Overall Error
Prediction on homo-stresses	11%	25.1%	241%	65.5%	48.1%	105.4%	82.7%
Prediction on true-stresses	11%	25.1%	39.2%	23.2%	13.1%	14.3%	21.1%

## Data Availability

Available upon request.

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
