# Peer review of "True Stress Theory of Matrix in A Composite: A Topical Review"

_materials, 2023, doi:10.3390/ma16020774_

Round 1
Reviewer 1 Report
It is good paper but needs a comprehensive verification step with the data published in the literature regarding mechanical properties of composites with and without notch.
· Composites show nonlinear stress stain behavior in matrix direction and in in-plane shear. so it is very important author clearly refer this point in the introduction. Following papers might help author in this regard. Polymer Reviews 60 (1), 42-85,2020; Polymer Testing 65, 44-53,2018; Journal of Reinforced Plastics and Composites 36 (3), 214-225,2017; Polymers and Polymer Composites 28 (3), 159-169, 2020; Composite Structures, Volume 292, 15 July 2022, 115671, 2022.
· The number of equations are more than 100, which should be reduced inside the paper. if it is necessary author can remove most of the unnecessary equations to the appendix.
· Predicted results by this method needs to be verified with the results published in the paper. This is a necessary step before the paper acceptance.
· Regarding SCF in composite materials and its effect on the strength of the composite materials, authors need to verify their model with the predictions by the researches and models ae Whitney and Nuismer and papers as: Compos. Mater. Eng 1 (1), 71-90, 2019; Advanced Composite Materials 27 (1), 67-83, 2018;
I did not mean for authors to refer to all the references I have mentioned in my report. I just asked them to read these papers to validate their research part regarding SCF and nonlinear modeling. So they are welcome to use any of my mentioned papers. If they find those papers unnecessary, they can use other papers in the literature to validate their research.
Author Response
Replies to the reviewers’ comments on Manuscript ID: materials-2093366
Reviewer #1
Comment: It is good paper but needs a comprehensive verification step with the data published in the literature regarding mechanical properties of composites with and without notch.
Reply: Many thanks for the reviewer to spend his time to review this manuscript. As already indicated in the title of the paper, this is a topical review paper, by summarizing the author’s works on the matrix true stress theory already published previously. Comprehensive verification and comparison with experiments have been shown in the author et al’s large number of previous publications. The latest papers include those in Comp. Sci. Tech., 207: 108665, 2021 and Comp. Sci. Tech., 219: 109206, 2022 for unnotched composites and in Comp. Struct., 306: 116558, 2023 for notched laminates. Moreover, at least 22 publications have been achieved by other researchers outside the author’s group who applied the matrix true stress theory in resolving their own composite failure problems. Those publications have been listed into the references of this paper. Verification must have been the first step before their applications.
Comment: Composites show nonlinear stress stain behavior in matrix direction and in in-plane shear. so it is very important author clearly refer this point in the introduction. Following papers might help author in this regard. Polymer Reviews 60 (1), 42-85,2020; Polymer Testing 65, 44-53,2018; Journal of Reinforced Plastics and Composites 36 (3), 214-225,2017; Polymers and Polymer Composites 28 (3), 159-169, 2020; Composite Structures, Volume 292, 15 July 2022, 115671, 2022.
Reply: This is a review paper on the matrix true stress theory, which is fundamental to predict properties of any composite outside a linear elasticity. It is not a research paper on any specific property of the composite. The influence of the matrix true stress theory on a composite nonlinear behavior is represented through two aspects. One is in that the stress components in an instantaneous compliance tensor of the matrix outside a linear elastic range must be true values, and the other is in that an interface debonding induced relative slippage between the fiber and matrix interfaces can only be analyzed based on the matrix true stress theory. In the revised manuscript, one of the reviewer’s mentioned references has been referred.
Comment: The number of equations are more than 100, which should be reduced inside the paper. if it is necessary author can remove most of the unnecessary equations to the appendix.
Reply: It is true that the numbering equations in this paper was somewhat different from a traditional way. No sub-numbering was carried out. It was only a different numbering way, and should not affect any understanding of a reader on the contents of this work. It is noted that all of those equations are necessary for the reader’s understanding. A major role of this work is to provide the Excel-table based running program to a reader to calculate all of the possible matrix SCFs in a composite. The program was written from the equations summarized in the paper.
Comment: Predicted results by this method needs to be verified with the results published in the paper. This is a necessary step before the paper acceptance.
Reply: As afore-mentioned, the purpose of this paper was to review and summarize a useful and newly established matrix true stress theory for obtaining various composite properties. The theory is consisted of 54 numbered equations, which has been derived and verified previously in many different publications. The present paper gathered them together and highlighted their applications to some typical areas in composite failures.
Comment: Regarding SCF in composite materials and its effect on the strength of the composite materials, authors need to verify their model with the predictions by the researches and models ae Whitney and Nuismer and papers as: Compos. Mater. Eng 1 (1), 71-90, 2019; Advanced Composite Materials 27 (1), 67-83, 2018;
Reply: The establishment of the matrix true stress theory and verification of it with various composite experiments, both by ourselves and by third parties have been done previously. Thanks for the reviewer’s suggestion, which will be considered elsewhere.
Comment: I did not mean for authors to refer to all the references I have mentioned in my report. I just asked them to read these papers to validate their research part regarding SCF and nonlinear modeling. So they are welcome to use any of my mentioned papers. If they find those papers unnecessary, they can use other papers in the literature to validate their research.
Rely: Many thanks for the reviewer’s comments.
Reviewer 2 Report
This review is about the homogenized stress of the matrix of a composite when the deformations are not reversible. The author reviewed the relations that he previously proposed. This review can be published after the following modifications:
1- The English of the manuscript is not fluent. Sometimes, the reader misses the point because of the ambiguous meaning of the phrases.
2- The quality of the figures can be modified.
3- Tables and equations are not correctly embedded into the text.
4- The Excel file can be developed a lot. Some instructions and descriptions can be added or creating an interactive window to insert the input by the user.
Author Response
Reviewer #2
Comment: This review is about the homogenized stress of the matrix of a composite when the deformations are not reversible. The author reviewed the relations that he previously proposed. This review can be published after the following modifications:
Reply: Thank you very much in spending your time and effort in reviewing this paper.
Comment: 1- The English of the manuscript is not fluent. Sometimes, the reader misses the point because of the ambiguous meaning of the phrases.
Reply: Thanks for the reviewer’s comment. The author paid most attention to the English in revising the manuscript, and a lot of sentences were rewritten. Hopefully a better understanding on the contents of the manuscript can be gained by a reader.
Comment: 2- The quality of the figures can be modified.
Reply: This comment will be better incorporated into the manuscript by the production editor before it is published.
Comment: 3- Tables and equations are not correctly embedded into the text.
Reply: Necessary changes have been made following the reviewer’s comment. It is noted that the author’s original manuscript was edited into the journal’s standard format by the journal’s editorial office, in which some equations seemed to have been less arranged into proper positions. These problems are believed to be resolved before publication.
Comment: 4- The Excel file can be developed a lot. Some instructions and descriptions can be added or creating an interactive window to insert the input by the user.
Reply: The input data for running the Excel file are quite limited, only those in blue color, and each of them has already been instructed in the file. The file can be run by a reader only when he/she has gone through this paper. Many SCFs can be calculated without iteration. However, to obtain three kinds of SCFs, some iterations are necessary. The Excel file is different from other computer language, e.g. Fortran, based program which can be written to a black-box, at least from the author’s knowledge. Before the reader uses the Excel file, he/she has to get familiar with the equations listed in the paper.

Reviewer 3 Report
Can be accepted. Paper is well written.
Author Response
Thank you!
Reviewer 4 Report
My comments for improvement of the manuscript,
*English is not good and should be revised deeply. There are many ambiguous texts.
*The equations numbering although correct, is irregular.
*There are many equations. I recommend transferring some of the unnecessary ones into the Appendix section.
*Tables 1 and 2 are mentioned wrongly within the text. If table 1 is the first table, it should be cited firstly in the text.
*Units in tables are unsuitable. For example in Table 2. There are 1.76 and 1.424. All the data should be in the same digits. Please write 1.760 and 1.424. This should be done in all tables.
*“Interface” in Equations 83-86 should be corrected. Its because the “MathType” software knows “Int” as a command for integral and separates it from the rest letters.
*Table 3 needs to be edited. Texts are overlapped.
*I do not know what is the purpose of this section: “Applications by other people”.
This section seems a literature review. The literature review should appear in the introduction in detail, not a short paragraph at the end of the paper. This doesn't seem right and is non-standard.
*The conclusion is not suitable. It should be associated with a brief definition of the work and the most important results.
*This sentence should be mentioned in the “Data availability” section,
“Analytical formulae for almost all of the true stress components have been derived by the author, and are summarized in the paper. They have been implemented into an Excel-table based program. If the reader has any question relevant to the Excel-table based program for calculating all of the matrix SCFs in a composite, he/she is welcome to write to the author at huangzm@tongji.edu.cn.”
*Self-citation of the author is unacceptable. 20 references out of 61!!. It should be reduced.
*Almost all the references are from one country. Not a fair referencing. This is an international journal, not an internal home one. Thus, references should be divided into other areas.
Author Response
Reviewer #4
Comment: *English is not good and should be revised deeply. There are many ambiguous texts.
Reply: Thanks for the reviewer’s time spent in reviewing this manuscript. The author has paid most attention to the English in revising the manuscript. Lots of sentences have been rewritten, and hopefully a better understanding can be gained by the reader
Comment: *The equations numbering although correct, is irregular.
Reply: Yes, the numbering on equations in this paper was different from a traditional way. No sub-numbering was carried out. It was only a different numbering way, and should not affect any understanding of a reader on the contents of this work.
Comment: *There are many equations. I recommend transferring some of the unnecessary ones into the Appendix section.
Reply: Yes, many equations are necessary for this work, but they are important. In general, only those which are already established or express detailed derivations for a theory are suitable to be arranged into an Appendix. This paper is of a topical review, without any derivation contained. More than half of the equations belong to the matrix true stress theory, which have been programmed into an Excel table file, whereas the others are also useful to the reader.
Comment: *Tables 1 and 2 are mentioned wrongly within the text. If table 1 is the first table, it should be cited firstly in the text.
Reply: Thanks for the comment. A change has been made accordingly in the revised manuscript.
Comment: *Units in tables are unsuitable. For example in Table 2. There are 1.76 and 1.424. All the data should be in the same digits. Please write 1.760 and 1.424. This should be done in all tables.
Reply: Necessary modifications have been made in the revised manuscript. It is noticed that the material property parameters shown in the present Table 2 (the previous Table 1) were copied from the references with no adjustment.
Comment: *“Interface” in Equations 83-86 should be corrected. Its because the “MathType” software knows “Int” as a command for integral and separates it from the rest letters.
Reply: Yes, indeed, the “Mathtype” software caused the problem, which will be corrected by the publisher in producing the manuscript.
Comment: *Table 3 needs to be edited. Texts are overlapped.
Reply: Done accordingly, and is believed to be further edited by the publisher.
Comment: *I do not know what is the purpose of this section: “Applications by other people”. This section seems a literature review. The literature review should appear in the introduction in detail, not a short paragraph at the end of the paper. This doesn't seem right and is non-standard.
Reply: The matrix true stress theory established by the author has been applied to resolving a lot of composite failures not only by the author’s group, but also by other researchers outside the author’s group. However, the purpose of this paper was not on detailed applications, thus only a brief mentioning together with a reference list for the other people’s applications was given in the paper. In the revised manuscript, necessary changes have been made to this section.
Comment: *The conclusion is not suitable. It should be associated with a brief definition of the work and the most important results.
Reply: Yes, the author agrees with the reviewer’s opinion. However, as the matrix true stress theory is completely new to the composite community, its role may be of the first and most concern by most readers. In the conclusion, the author first emphasized the role of this theory, followed by the definition and the main purpose as well as most important part of this work.
Comment: *This sentence should be mentioned in the “Data availability” section,
“Analytical formulae for almost all of the true stress components have been derived by the author, and are summarized in the paper. They have been implemented into an Excel-table based program. If the reader has any question relevant to the Excel-table based program for calculating all of the matrix SCFs in a composite, he/she is welcome to write to the author at huangzm@tongji.edu.cn.”
Reply: The author only provided brief information at the end of the paper to let the reader to freely contact him if necessary. No need to add a special “Data availability” section, since at least more words will be added to the paper. If, however, the publication format will require this section, the reviewer’s comment will be followed accordingly.
Comment: *Self-citation of the author is unacceptable. 20 references out of 61!!. It should be reduced.
*Almost all the references are from one country. Not a fair referencing. This is an international journal, not an internal home one. Thus, references should be divided into other areas.
Reply: This is a review and summary paper of the previous publications on the matrix true stress theory. This theory has been established and nearly matured only by the author and his group. There is indeed no other publication relevant to the development of the theory yet. A total number of 22 publications by the other people both in home and abroad have dealt with applications of the matrix true stress theory to resolving different composite failures. They have been listed into the references of this paper.
Round 2
Reviewer 1 Report
accept
Reviewer 2 Report
The author has answered the comments and modified the manuscript. The paper can be considered for possible publication after the English check.